# Representation of time interval entrained by periodic stimuli in the visual thalamus of pigeons

Yan Yang[†], Qian Wang[†], Shu-Rong Wang, Yi Wang*, Qian Xiao*

State Key Laboratory of Brain and Cognitive Sciences, Institute of Biophysics, Chinese Academy of Sciences, Beijing, China

**Abstract** Animals use the temporal information from previously experienced periodic events to instruct their future behaviors. The retina and cortex are involved in such behavior, but it remains largely unknown how the thalamus, transferring visual information from the retina to the cortex, processes the periodic temporal patterns. Here we report that the luminance cells in the nucleus dorsolateralis anterior thalami (DLA) of pigeons exhibited oscillatory activities in a temporal pattern identical to the rhythmic luminance changes of repetitive light/dark (LD) stimuli with durations in the seconds-to-minutes range. Particularly, after LD stimulation, the DLA cells retained the entrained oscillatory activities with an interval closely matching the duration of the LD cycle. Furthermore, the post-stimulus oscillatory activities of the DLA cells were sustained without feedback inputs from the pallium (equivalent to the mammalian cortex). Our study suggests that the experience-dependent representation of time interval in the brain might not be confined to the pallial/cortical level, but may occur as early as at the thalamic level.
DOI: https://doi.org/10.7554/eLife.27995.001

*For correspondence:
yiwang@ibp.ac.cn (YW);
qianxiao@moon.ibp.ac.cn (QX)

[†]These authors contributed equally to this work

Competing interests: The authors declare that no competing interests exist.

## Introduction

The ability to process and sense the temporal information of external stimuli is fundamental for humans and other animals in enabling them to adapt to the environment. Use of the perceived temporal information from previously experienced stimuli to predict upcoming events has been demonstrated in birds (*Gibbon et al., 1984*; *Henderson et al., 2006*; *Kalenscher et al., 2006*), rodents (*Crystal and Baramidze, 2007*; *Agostino et al., 2011*), primates, and humans (*Leon and Shadlen, 2003*; *Buhusi and Meck, 2005*; *Janssen and Shadlen, 2005*; *Penney et al., 2008*).

Previous studies on the neural mechanism underlying time perception and representation have mainly focused on the cerebellum (*Malapani et al., 1998*; *Harrington et al., 2004*), striatum (*Meck et al., 2008*; *Mello et al., 2015*), and cortex of humans and mammals (*Leon and Shadlen, 2003*; *Xu et al., 2014*). A recent study showed, however, that some retinal ganglion cells in salamander and mouse retinae signal the time at which an omitted stimulus in a sequence of flashes would occur (*Schwartz et al., 2007*). There is also increasing evidence showing that the visual thalamus filters, rather than passively relays, the visual information sensed by the retina as it signals to the cortex (*Cudeiro and Sillito, 2006*; *Saalmann and Kastner, 2011*; *Sherman, 2016*). Thus, the thalamus may also be involved in coding the temporal information of periodic visual stimuli. However, we have limited knowledge on how the visual thalamus encodes the temporal information from experienced stimuli and represents the time that has elapsed since the previous event.

To address this question, we used the pigeon as an animal model and the nucleus dorsolateralis anterior thalami (DLA) as the target brain area. The pigeon is a common animal model for behavioral and neurobiological studies of time-dependent cognitive tasks, such as interval timing (*Kalenscher et al., 2006*), sequence learning (*Helduser et al., 2013*; *Lissek et al., 2013*), and

**eLife digest** Being able to track the passage of time enables animals to predict when events will occur in the future. This in turn helps them optimize their behavior. Hummingbirds, for example, schedule their foraging visits so that they return to each flower after it has had time to replenish its supply of nectar. In the laboratory, rats and monkeys can learn to delay their responses to a cue until a specific period of time has elapsed in order to earn a reward. But how does the brain keep track of time?

To discover what is happening in the world around us, we rely on our eyes and other sense organs to collect sensory information. These organs send this information to the thalamus, a structure deep below the surface of the brain. The thalamus processes and filters the sensory information, and then forwards it to the cortex located in the brain's outer layer. Experiments have shown that, after training with events that occur at regular intervals, cells in the cortex and the eye can signal the time at which the next event would occur. But it was not known if cells in the thalamus could do this too.

To answer this question, Yang, Wang et al. recorded from the thalamus of pigeons while exposing the birds to alternating periods of light and darkness. Pigeons were chosen because they have good eyesight and perform well on time-tracking tasks. A set of cells in the pigeon thalamus changed their activity levels to follow each light/dark switch. The cells tracked switches that occurred every few minutes as accurately as those that occurred every second. Next, Yang, Wang et al. stopped switching the light on and off, and instead left the light either on or off for 2-3 hr. Even in constant light or darkness during the 2-3 hr, some of the cells maintained their previous pattern of firing. In other words, the cells continued to signal the time when the light/dark switches should occur long after the switching had been stopped. Inactivating the pigeon's equivalent area of the mammalian cortex in the brain had no effect on this response.

The findings of Yang, Wang et al. suggest that the thalamus – like the cortex and the eye – can track events that occur at regular intervals, at least in pigeons. The next step is to determine whether the thalamus encodes time intervals in other species as well, and how this might help the animals to optimize time-dependent behaviors, such as foraging and navigation.
DOI: https://doi.org/10.7554/eLife.27995.002

delayed matching-to-sample (*Browning et al., 2011*). Furthermore, to achieve foraging and safe flight successfully in diverse environments, birds have developed a complex visual system that is superior to that of most vertebrates (*Shimizu and Watanabe, 2012*; *Wylie et al., 2015*). The avian DLA receives direct retinal inputs, projects onto the pallial Wulst in both hemispheres (*Karten et al., 1973*; *Bagnoli and Burkhalter, 1983*; *Miceli et al., 1987, 2008*), and receives feedback inputs from the pallial Wulst (*Karten et al., 1973*; *Miceli et al., 1987*). The avian retina-DLA-Wulst pathway is comparable to the mammalian retina-lateral geniculate nucleus (LGN)-striate visual pathway (*Shimizu and Karten, 1993*). The avian Wulst further projects onto the nidopallium caudolaterale (NCL), which is comparable to the mammalian prefrontal cortex (PFC) (*Kröner and Güntürkün, 1999*) (*Figure 1A*). More importantly, the luminance cells in the pigeon DLA can encode ambient luminance (*Yang et al., 2005*). During visual conditioning, the responses of the DLA luminance cells to the conditioned stimulus (CS: whole-field light) are modified by training, and the training-induced changes that occur in response to the CS are in parallel with the acquisition of the behavioral responses of pigeons (*Gibbs et al., 1986*).

Using electrophysiological single-unit recordings, we compared the neuronal responses of DLA luminance cells before, during, and after the repetitive presentation of light/dark (LD) stimuli with intervals ranging from seconds to minutes (L/D: 1 s/1 s to 240 s/240 s, 5 to 25 cycles). All luminance cells had steady firing rates under constant photic conditions and synchronized their activities with the rhythmic luminance changes of LD stimuli. After LD stimulation, some luminance cells retained the entrained oscillatory activities even when the photic conditions were constant. The post-stimulus replay responses of these cells were dependent on the time interval and number of LD cycles of periodic stimuli applied during LD stimulation. Both Wulst pharmacological inactivation and electrolytic lesions did not affect the post-stimulus oscillatory responses of DLA cells entrained by the periodic

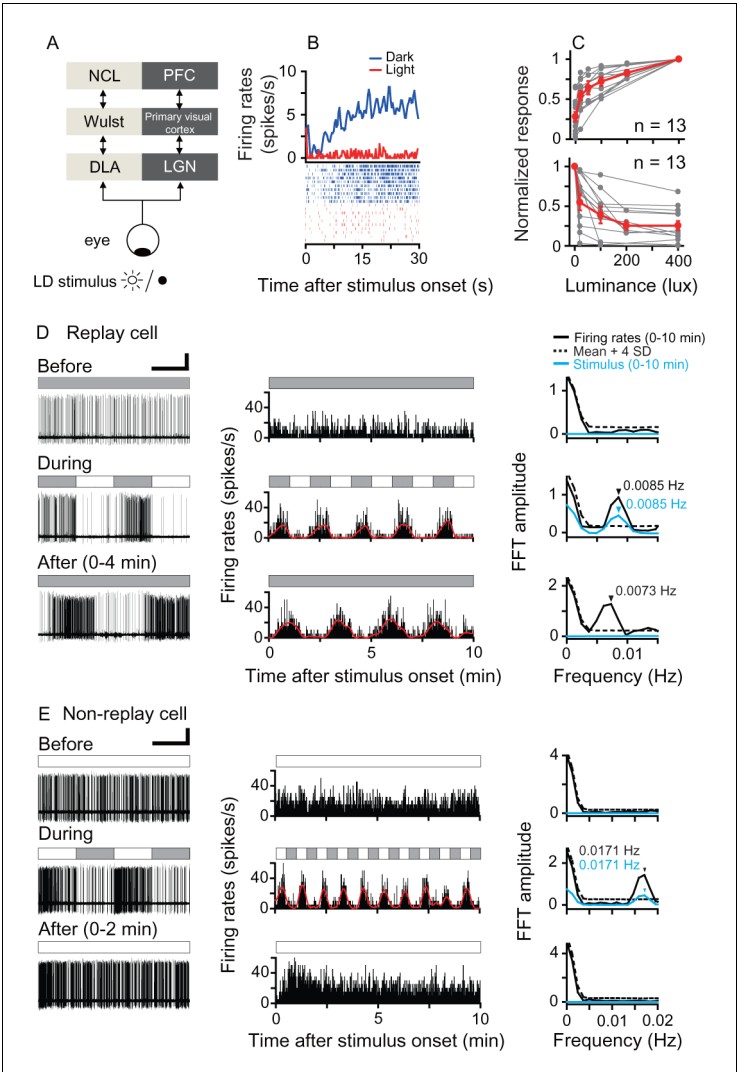

**Figure 1.** Responses of luminance cells in the pigeon DLA to periodic light/dark (LD) stimuli. (**A**) The avian nucleus dorsolateralis anterior thalami (DLA) receives direct retinal inputs and projects to the pallial Wulst, which further projects to the nidopallium caudolaterale (NCL). The DLA, Wulst, and NCL in birds (light gray) are comparable to the lateral geniculate nucleus (LGN), primary visual cortex, and prefrontal cortex (PFC) in mammals (dark gray), respectively. (**B**) Histogram (top, bin = 0.5 s) and raster (bottom, 10 repeats) showing the response of a light-suppressed cell to the LD stimulus (light: 200 lux; dark: 0 lux). (**C**) Normalized responses of light-activated cells (top) increased, whereas those of light-suppressed cells (bottom) decreased when the luminance level was raised in steps. The responses of each cell to each luminance level were averaged across 5–10 repeats. The luminance responses of each cell were normalized by the cell's maximal response to all tested luminance levels (gray symbols). Mean values (± SEM) across all cells are presented by red symbols. (**D, E**) Original recording traces (left column), response histograms (middle column, bin = 0.2 s), and corresponding Fast Fourier Transformation (FFT) analyses (right column) before, during, and after LD stimulation were plotted for a typical DLA replay cell (**D**) and a non-replay cell (**E**). Filled and empty blocks in the horizontal bars indicate dark (0 lux) and light (200 lux) photic conditions. For a clear illustration of firing pattern changes, the response histogram of each cell was filtered by a zero-phase, low-pass Butterworth filter. The filtered histogram is marked with a red line. As shown in the right column, the oscillation frequency of the response histogram of the recorded cell (black line) was significant when its FFT amplitude was higher than the mean +4 SDs of the control (dotted line) estimated by 1000 bootstrap re-sampling of the original response histogram. The oscillation frequency of the photic stimulus is marked with a blue line. Black and blue arrows indicate significant oscillation frequencies of the response histograms and photic stimulus, respectively. Scale bars: 60 s, 50 µV and 30 s, 50 µV in left columns of (**D**) and (**E**), respectively.

DOI: https://doi.org/10.7554/eLife.27995.003

The following source data and figure supplement are available for figure 1:

*Figure 1 continued on next page*

*Figure 1 continued*

**Source data 1.** Luminance responses of DLA cells to periodic photic stimuli.
DOI: https://doi.org/10.7554/eLife.27995.005
**Figure supplement 1.** Histological identification and distribution of recording sites marked with electrolytic lesions.
DOI: https://doi.org/10.7554/eLife.27995.004

LD stimuli, suggesting that the intrinsic circuits in the DLA play the primary role in representing the time interval of periodic events experienced previously.

## Results

### Photic responses of DLA luminance cells were modified by periodic LD stimuli experienced previously

We recorded 190 DLA luminance cells from 19 animals (*Figure 1—figure supplement 1*). These cells encoded ambient luminance and sustained steady activities under constant photic conditions. Their firing rates either increased (light-activated) or decreased (light-suppressed) monotonically when the luminance level of the stimulus was raised in steps (*Figure 1B,C*). During LD stimulation, all luminance cells synchronized their firing rates with the rhythmic luminance changes of LD stimuli whose temporal frequencies ranged from 0.5 Hz (L/D: 1 s/1 s) to 0.002 Hz (L/D: 240 s/240 s). The oscillation frequencies of the entrained activities of these cells were closely correlated with the temporal frequencies of the luminance changes of LD stimuli (linear regression, slope = 0.99, $R^2$ = 0.99). After LD stimulation, the light-activated cells were tested under light, whereas the light-suppressed cells were tested in darkness. Neuronal activities of each luminance cell after LD stimulation were continuously recorded for 2–3 hr. We found that 54 cells (54/190 = 28%) retained the entrained oscillatory activities under constant photic conditions after LD stimulation. Thus, these cells were referred to as replay cells, which included 25 light-activated and 29 light-suppressed cells. Before LD stimulation, a typical DLA replay cell exhibited steady spontaneous activity in constant darkness (*Figure 1D*, top row). During LD stimulation, the firing rates of this typical DLA replay cell oscillated at the same frequency (0.0085 Hz) as the luminance changes of the LD stimulus (L/D: 60 s/60 s) (middle row). As shown in the response histogram, in the first 10 min after 10 LD cycles (bottom row), the neuronal activity of this cell continued to oscillate at almost the same frequency (0.0073 Hz) as the LD stimulus.

The remaining DLA cells (136/190 = 72%) displayed non-oscillatory response patterns after LD stimulation and were therefore referred to as non-replay cells. For example, a typical DLA non-replay cell (*Figure 1E*) had steady excitatory responses in constant light (200 lux). This cell synchronized its firing rates with the luminance changes of the periodic photic stimulus (L/D: 30 s/30 s). After 25 LD cycles, this cell first had excitatory responses to the light onset and then returned to a steady firing pattern, rather than continuing the oscillatory activity under constant light.

### Post-stimulus activities of replay cells were correlated with parameters of periodic stimuli

The neuronal responses of replay cells after LD stimulation were regulated by both the temporal frequency of luminance changes and the number of LD cycles during LD stimulation. To evaluate the effects of stimulus parameters on the neuronal activities of replay cells after LD stimulation, the temporal frequency of LD stimuli was varied from 0.5 Hz (L/D: 1 s/1 s) to 0.002 Hz (L/D: 240 s/240 s) and the number of LD cycles was varied within a range from 5 to 25 cycles. We found that, first, replay cells did not exhibit oscillatory activities after LD stimulation when the duration of the LD cycle was shorter than 10 s (L/D: 5 s/5 s). The oscillation frequencies of replay cells (n = 54 cells) after LD stimulation were highly correlated with the temporal frequencies of LD stimuli (slope = 0.62, $R^2$ = 0.99) (*Figure 2A*). Furthermore, once a replay cell returned to its steady non-oscillatory activity under constant photic conditions, it could be re-entrained by visual stimuli of other frequencies (6 cells, *Figure 2B*). The duration of the first replay cycle after LD stimulation was also linearly correlated with the duration of the LD cycle (slope = 0.58, $R^2$ = 0.95, n = 54 cells) (*Figure 2C*). Second, the

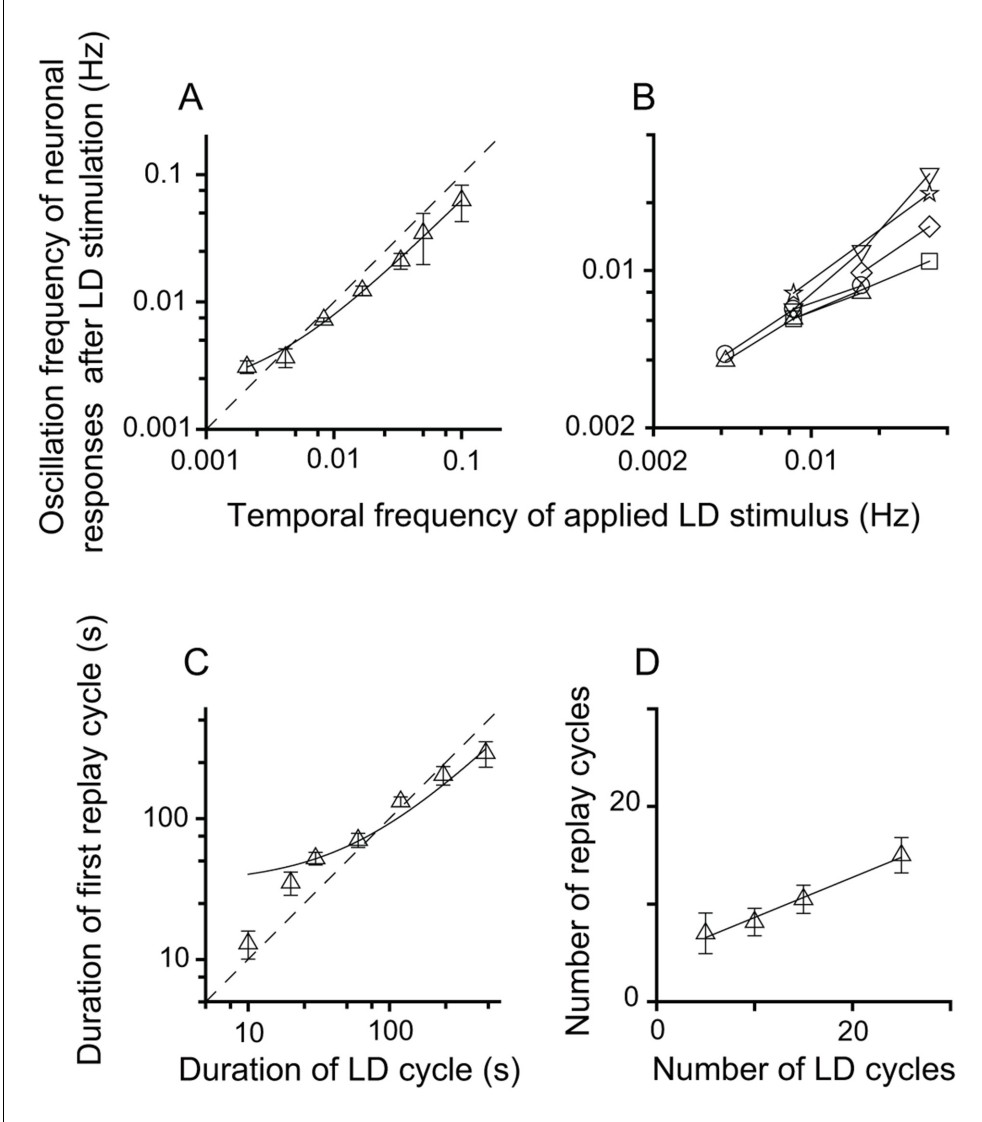

**Figure 2.** Relationship between parameters of the LD stimulus and neuronal responses of the replay cells after LD stimulation in the DLA. (**A**) The oscillation frequencies of the responses of DLA replay cells (mean ± SEM, n = 54 cells) in the first 14 min after LD stimulation were linearly correlated with the temporal frequencies of LD stimuli (log-log scale). (**B**) Replay cells (n = 6 cells) could be re-entrained by periodic stimuli of different temporal frequencies (log-log scale). Each type of symbol represents a replay cell. (**C**) Duration of the first replay cycle of replay cells (mean ± SEM, n = 54 cells) was closely related to the duration of the LD cycle during LD stimulation (log-log scale). (**D**) The number of replay cycles (mean ± SEM, n = 21 cells) monotonically increased when the number of LD cycles (L/D: 30 s/30 s) increased. The dashed lines in (**A**) and (**C**) are the diagonal lines. The data in (**A**), (**C**), and (**D**) are fitted by a linear function.

DOI: https://doi.org/10.7554/eLife.27995.006

The following source data is available for figure 2:

**Source data 1.** Oscillatory responses of DLA replay cells after periodic photic stimulation.
DOI: https://doi.org/10.7554/eLife.27995.007

increase in the number of LD cycles led to an increase in the oscillation time of replay cells after LD stimulation. When the temporal frequency of the LD stimulus was set to 0.0165 Hz (L/D: 30 s/30 s), the number of replay cycles increased when the number of LD cycles increased (slope = 0.41, $R^2$ = 0.99, n = 21 cells) (**Figure 2D**).

## Post-stimulus oscillatory activities of replay cells declined with time

After LD stimulation, the entrained oscillatory activities of replay cells gradually returned to non-oscillatory activities with time. The change was reflected in the decrease of oscillation frequency and response amplitude. *Figure 3* shows the whole neuronal response process of a replay cell before, during, and after LD stimulation. This replay cell had steady non-oscillatory activity in darkness before LD stimulation and showed oscillatory activity (0.0085 Hz) induced by LD stimuli (L/D: 60 s/60 s). After 10 LD cycles, the entrained oscillatory activity of this cell continued for up to 16 replay cycles (~48 min) in darkness. The oscillation frequency of the post-stimulus responses gradually slowed down over time (*Figure 3A,B*). In the first 7 min after LD stimulation, this cell displayed 0.0073 Hz oscillation, close to the temporal frequency of the LD stimulus (0.0085 Hz). As time elapsed, the oscillation frequency of this cell decreased to 0.0061 Hz in the period 19–26 min and to 0.0049 Hz in the period 38–45 min after withdrawing the LD stimulus. In addition, the ratio of the duration of the first replay cycle to the duration of the LD cycle (120 s) was 1.24, which increased to 2.21 in the last replay cycle (*Figure 3C*). The peak activity of each replay cycle was divided by the mean response (37.19 spikes/s) averaged across all dark periods during LD stimulation. The normalized peak activity of this cell decreased from 1.09 in the first replay cycle to 0.43 in the last replay cycle (*Figure 3E*).

To further examine the firing pattern changes of replay cells over time, we chose the first three replay cycles after LD stimulation and the last three replay cycles before returning to non-oscillatory activities. We observed that the duration of the replay cycle slowly increased over time (one-way ANOVA, $F_{5, 180} = 4.38$, $p < 0.001$, n = 31 cells; *Figure 3D*). For the same group of replay cells, the peak activity of the replay cycle gradually decreased over time ($F_{5, 180} = 3.59$, $p = 0.004$) (*Figure 3F*).

## Post-stimulus oscillatory activities of replay cells retained after pharmacological inactivation or electrolytic lesions of the Wulst in both hemispheres

The avian DLA has reciprocal connections with the pallial Wulst. To clarify the possible effect of the inputs from Wulst on the DLA replay cells, the Wulst in both hemispheres was temporarily inactivated by multi-site muscimol injections (*Figure 4—figure supplement 1*). We recorded 12 replay cells from 10 animals, and then compared their post-stimulus activities before and after Wulst inactivation. The tested replay cells included five light-activated and seven light-suppressed cells.

Wulst inactivation attenuated the spontaneous activities of replay cells, but had no effect on their photic responses. Under constant photic conditions before LD stimulation, the spontaneous firing rates of replay cells were slightly reduced from $3.89 \pm 0.87$ spikes/s (mean ± SEM) before injection to $2.3 \pm 0.51$ spikes/s after injection (paired t-test, $t_{pre\ vs.\ post} = 2.37$, $p = 0.03$, n = 12 cells). Furthermore, the photic responses of the DLA cells were evaluated by a preference index of their photic responses (R) to light (L = 200 lux) and dark stimuli (D = 0 lux) (index = $(R_L - R_D) / (R_L + R_D)$). No significant change in the preference index was observed after Wulst inactivation for the light-activated cells (pre-injection: $0.34 \pm 0.15$; post-injection: $0.5 \pm 0.22$; Wilcoxon rank sum test, ranksum = 21, $p = 0.22$, n = 5 cells) or for light-suppressed cells (pre-injection: $-0.52 \pm -0.19$; post-injection: $-0.43 \pm -0.16$; ranksum = 46, $p = 0.45$, n = 7 cells).

The entrained post-stimulus oscillatory activities of replay cells were still sustained when the Wulst was temporarily inactivated, as shown for a single replay cell (*Figure 4—figure supplement 2*). Neither the linear correlation between the post-stimulus oscillation frequencies of replay cells and the temporal frequencies of LD stimuli (pre-injection: slope = 0.62, $R^2 = 0.92$; post-injection: slope = 0.56, $R^2 = 0.82$; *Figure 4A*) nor the linear correlation between the duration of the first replay cycle and the duration of the LD cycle were significantly affected by Wulst inactivation (pre-injection: slope = 0.59, $R^2 = 0.91$; post-injection: slope = 0.73, $R^2 = 0.96$; *Figure 4C*). After Wulst inactivation, the replay cells could still be re-entrained by periodic stimuli of different frequencies (n = 4 cells, *Figure 4B*). The number of replay cycles increased as the number of LD cycles (30 s/30 s) increased (pre-injection: slope = 0.62, $R^2 = 0.68$; post-injection: slope = 0.54, $R^2 = 0.64$, n = 10 cells) (*Figure 4D*). In addition, for the same periodic stimulus, the post-stimulus activities of replay cells before and after injection showed no significant differences in oscillation frequencies (paired t-test, $t_{pre\ vs.\ post} = 0.32$, $p = 0.75$ ; *Figure 4A*), duration of the first replay cycle ($t_{pre\ vs.\ post} = -1.71$,

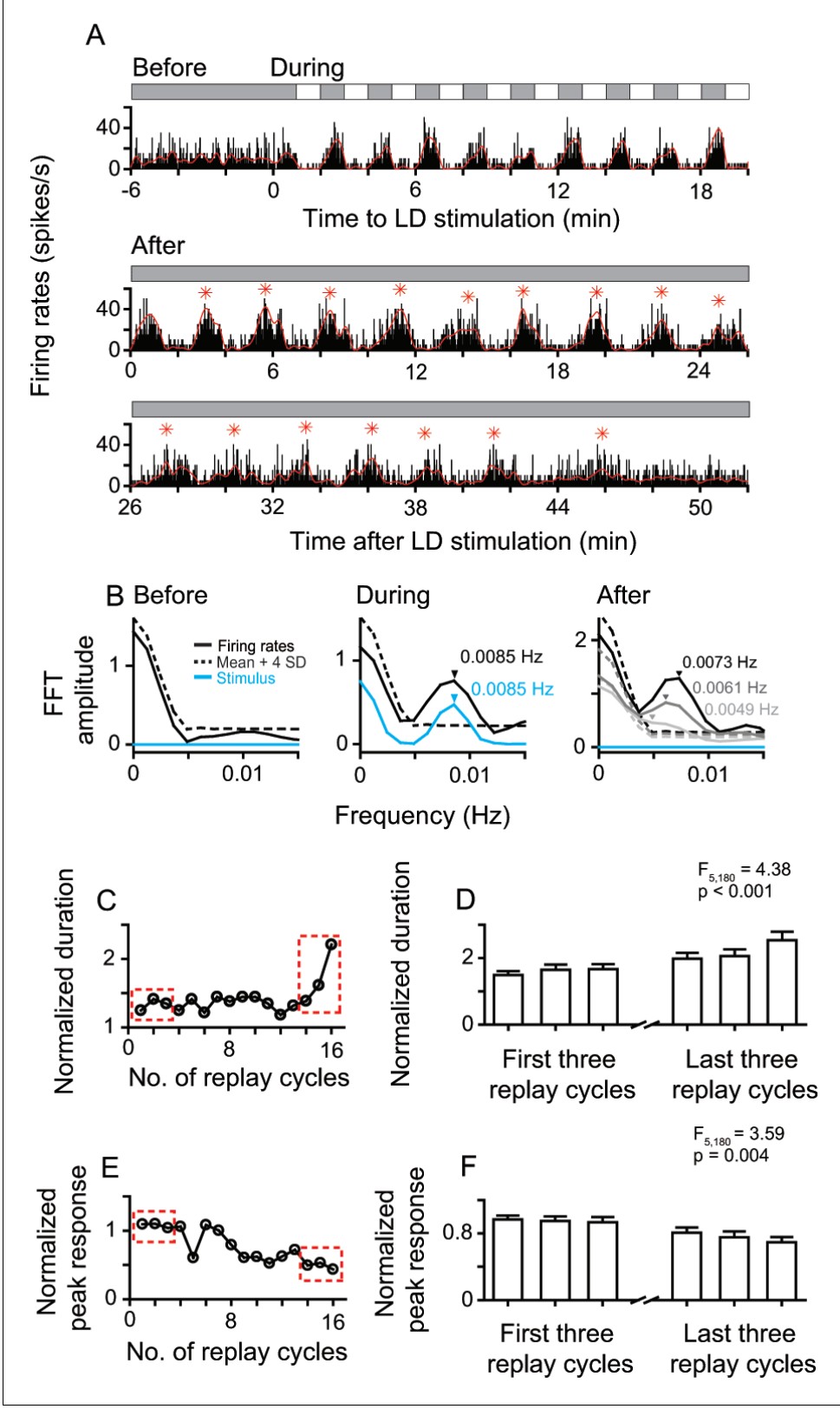

**Figure 3.** Sustained and attenuated oscillatory activities of replay cells after LD stimulation over time. (**A**) A typical replay cell had steady non-oscillatory activity in constant darkness, and synchronized its firing rates with luminance changes during LD stimulation (L/D: 60 s/60 s, 10 cycles, bin = 0.2 s). After LD stimulation, the entrained oscillatory

*Figure 3 continued on next page*

*Figure 3 continued*

activity of this cell was maintained for ~48 min and then returned to non-oscillatory activity. Red asterisks indicate the peak response time of replay cycles. (B) FFT analyses of response histograms of this cell before, during, and after LD stimulation. In the right panel, black: 0–7 min; dark gray: 19–26 min: light gray 38–45 min after LD stimulation. (C, E) Duration (C) and peak response (E) of each replay cycle of this cell. (D, F) Mean durations (D) and peak responses (F) of 31 replay cells during the first and last three replay cycles. Error bars represent SEM. For other conventions, see *Figure 1*.

DOI: https://doi.org/10.7554/eLife.27995.008

The following source data is available for figure 3:

**Source data 1.** Sustained and attenuated oscillatory responses of replay cells after periodic stimulation.
DOI: https://doi.org/10.7554/eLife.27995.009

---

p = 0.1; *Figure 4C*), or number of replay cycles ($t_{pre\ vs.\ post}$ = –0.94, p = 0.37; *Figure 4D*). Furthermore, the post-stimulus activities of replay cells declined over time before and after injection, which was reflected in the increasing duration of replay cycles (one-way ANOVA, pre-injection: $F_{5,\ 66}$ = 4.05, p = 0.002; post-injection: $F_{5,\ 66}$ = 4.55, p < 0.0001, n = 12 cells; *Figure 4E*) and decreasing peak responses of replay cycles over time (pre-injection: $F_{5,\ 66}$ = 3.17, p = 0.01; post-injection: $F_{5,\ 66}$ = 2.94, p = 0.01; *Figure 4F*).

We compared the spontaneous activities of Wulst cells before and after muscimol injection to prove that muscimol could effectively inhibit these cells. To examine the spatial range and temporal course of the muscimol inhibitory effect in the Wulst, Wulst cell activities were determined by the mean firing rates of eight recording positions. Each recording site was 0.5 mm from the injection site (*Figure 4—figure supplement 3*). We measured the spontaneous activity of Wulst cells averaged for eight recording sites surrounding the injection site before and after injection. Before the injection, the spontaneous firing rates of the Wulst cells surrounding the injection sites were 6.55 ± 0.85 spikes/s (n = 3 injection sites). Their firing rates between 0.25 hr and 1.5 hr after injection were only 15.9% ± 3.7% (1.08 ± 0.37 spikes/s) of those before injection, and slowly increased to 26.3% ± 0.3% (1.72 ± 0.22 spikes/s) between 1.5 hr and 3 hr after injection. Taken together, these data illustrate that muscimol effectively inhibited the Wulst cells, but that successful Wulst inactivation did not affect the entrained post-stimulus oscillatory activities of the DLA replay cells.

The pharmacological inactivation of the pallial Wulst had no significant effect on the post-stimulus oscillatory activities of the DLA replay cells. To exclude the possible impact of incomplete Wulst inactivation, electrolytic lesions were applied in the Wulst of both hemispheres (*Figure 5—figure supplement 1*).

We recorded 29 replay cells from nine Wulst-lesioned animals, which included 14 light-activated and 15 light-suppressed cells. By comparing the neuronal responses of the DLA replay cells in normal and Wulst-lesioned animals, we found that Wulst lesions did not affect the spontaneous activities or photic responses of the DLA replay cells. Under constant photic conditions before LD stimulation, no significant firing rate changes were observed between the replay cells recorded in the normal animals (3.39 ± 0.43 spikes/s) and those in Wulst-lesioned animals (3.54 ± 0.36 spikes/s) (t-test, $t_{normal\ vs.\ lesion}$ = –0.26, p = 0.39, $n_{normal}$ = 54 cells, $n_{lesion}$ = 29 cells). Furthermore, the photic responses of the DLA replay cells were not affected by Wulst lesions, which was reflected in the comparable preference indices of light-activated cells (normal animals: 0.46 ± 0.06, n = 25 cells; lesion animals: 0.5 ± 0.05, n = 14 cells, t-test, $t_{normal\ vs.\ lesion}$ = –0.46, p = 0.64) and those of light-suppressed cells (normal animals: −0.52 ± −0.06, n = 29 cells; lesion animals: −0.42 ± −0.03, n = 15 cells, $t_{normal\ vs.\ lesion}$ = -1.38, p = 0.17).

The post-stimulus oscillatory activities of replay cells were still retained after the application of Wulst lesions in both hemispheres. After LD stimulation, the replay cells still showed oscillatory activities, and the oscillation frequencies were linearly correlated with the temporal frequencies of LD stimuli (slope = 0.62, $R^2$ = 0.99, n = 29 cells) (*Figure 5A*). Furthermore, replay cells (six cells) could be re-entrained by periodic stimuli of different frequencies (*Figure 5B*). The duration of the first replay cycle of these cells was linearly correlated with that of the LD cycle (slope = 0.58, $R^2$ = 0.96, *Figure 5C*). The number of replay cycles increased when the number of LD cycles (L/D: 30 s/30 s) increased (slope = 0.86, $R^2$ = 0.94, n = 20 cells; *Figure 5D*). Furthermore, the post-stimulus oscillatory activities of the replay cells declined with time. The duration of the replay cycles increased

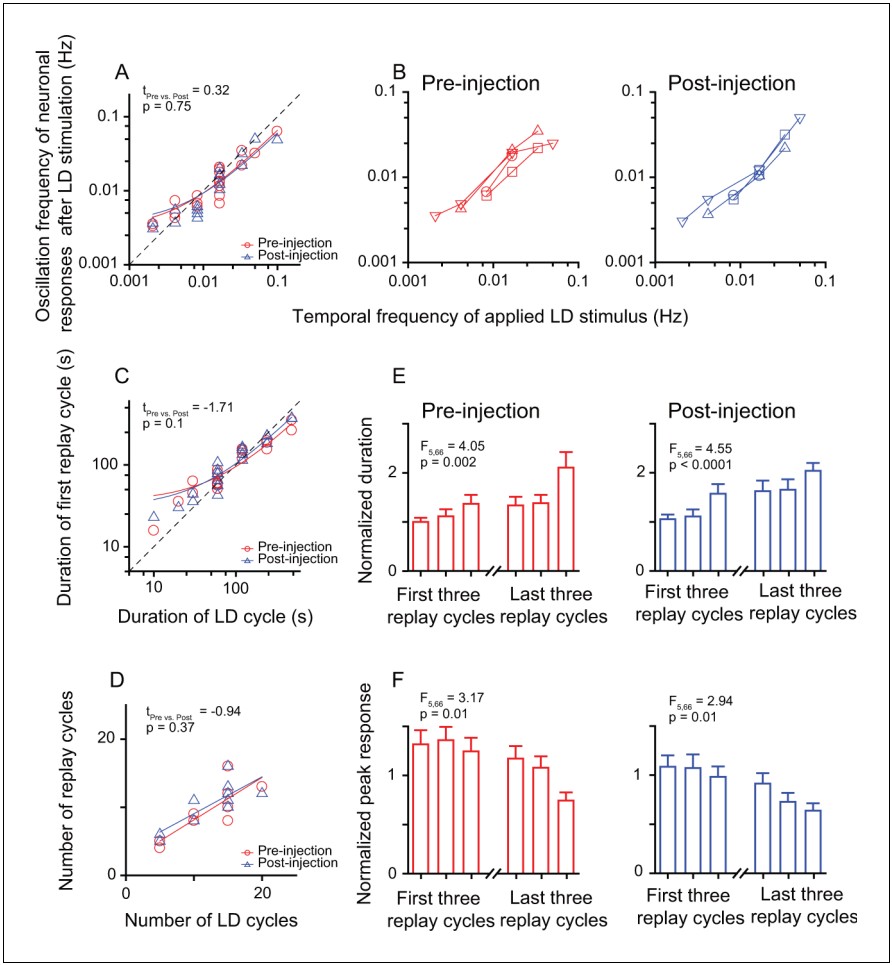

**Figure 4.** Inactivation of the Wulst had no effect on the entrained oscillatory responses of DLA replay cells after periodic stimulation. (**A**) Correlation between the oscillation frequencies of replay cells (n = 12 cells) in the first 14 min after LD stimulation and the temporal frequencies of LD stimuli (log-log scale). (**B**) A single replay cell could be re-entrained by periodic stimuli of different temporal frequencies (n = 4 cells, log-log scale) before (left) and after Wulst inactivation (right). Each type of symbol represents a replay cell. (**C**) Correlation between the duration of the first replay cycle of the replay cells and the duration of the LD cycle (log-log scale) (n = 12 cells). (**D**) Number of replay cycles monotonically increased when the number of LD cycles (L/D: 30 s/30 s) increased (n = 10 cells). (**E**, **F**) Mean durations (**E**) and mean peak responses (**F**) of the first and last three replay cycles of the replay cells (n = 12 cells) before (left) and after Wulst inactivation (right). The dashed lines in (**A**) and (**C**) are the diagonal lines. The data in (**A**), (**C**), and (**D**) are fitted by a linear function. Error bars in (**E**) and (**F**) represent SEM. For other conventions, see *Figures 2* and *3*.

DOI: https://doi.org/10.7554/eLife.27995.010

The following source data and figure supplements are available for figure 4:

**Source data 1.** Post-stimulus responses of replay cells before and after Wulst inactivation in both hemispheres.
DOI: https://doi.org/10.7554/eLife.27995.014
**Figure supplement 1.** Location of injection sites of bilateral Wulst inactivation in the pigeon brain.
DOI: https://doi.org/10.7554/eLife.27995.011
**Figure supplement 2.** Neuronal responses of a single DLA replay cell before and after Wulst inactivation in both hemispheres.
DOI: https://doi.org/10.7554/eLife.27995.012
**Figure supplement 3.** Inhibition of neuronal activities in the Wulst by muscimol.
DOI: https://doi.org/10.7554/eLife.27995.013

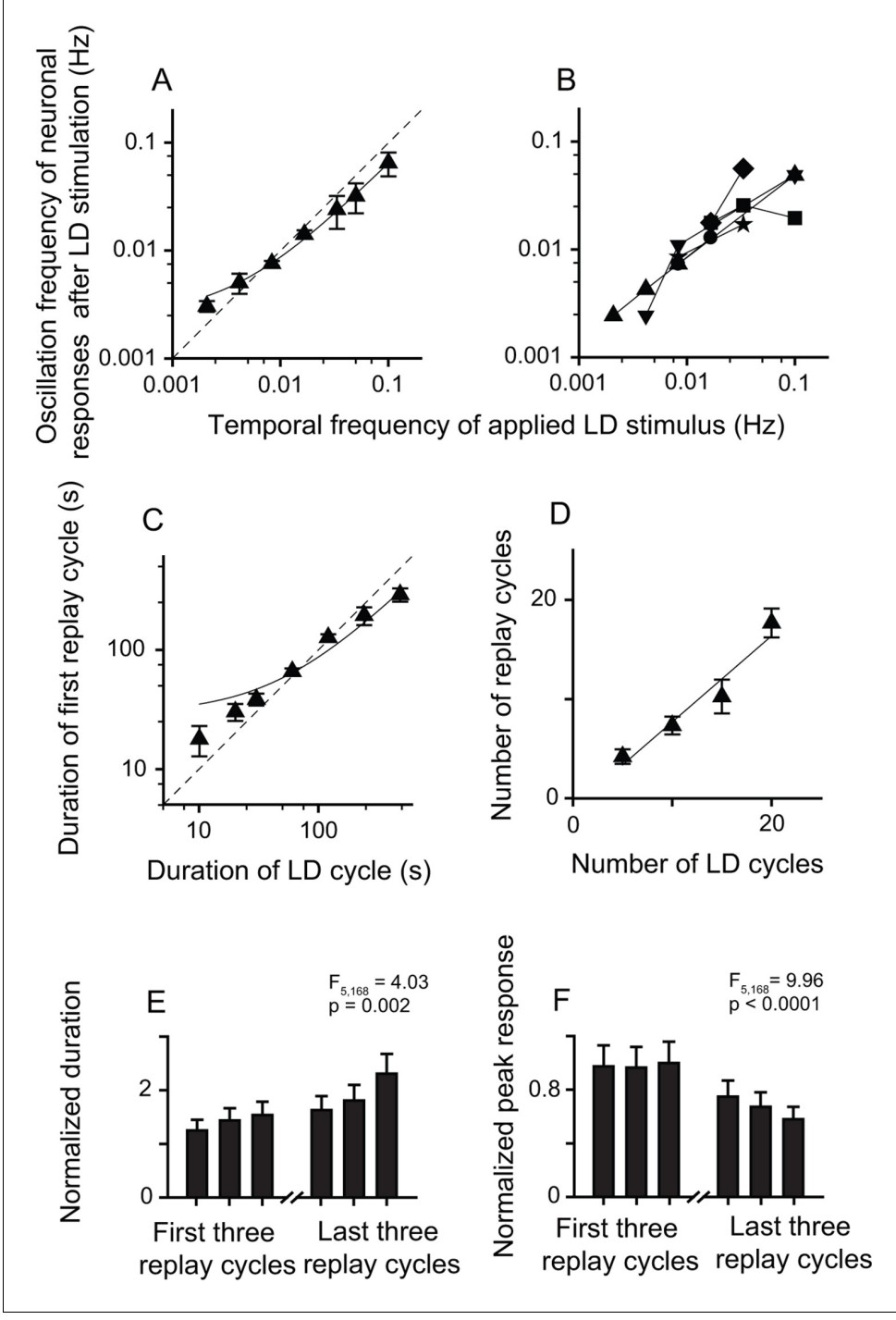

**Figure 5.** Bilateral Wulst lesions had no effect on the entrained oscillatory responses of the replay cells after periodic LD stimulation. (A) Oscillation frequencies (mean ± SEM) of 29 cells in the first 14 min after LD stimulation were linearly correlated with the temporal frequencies of LD stimuli (log-log scale). (B) A single replay cell could be re-entrained by periodic stimuli of different temporal frequencies (n = 6 cells, log-log scale). Each type of symbol represents a replay cell. (C) Duration of the first replay cycle of the replay cells (mean ± SEM, n = 29 cells) was closely related to the duration of the LD cycle during LD stimulation. (D) Number of replay cycles (mean ± SEM, n = 20 cells) monotonically increased when the number of LD cycles (L/D: 30 s/30 s) increased. (E, F) Mean durations (E) and peak responses (F) of the first and last three replay cycles of the replay cells (n = 29 cells). Dashed lines in (A) and (C) are the diagonal lines. Data in (A), (C), and (D) are fitted by the linear function. For other conventions, see *Figures 2* and *3*.

*Figure 5 continued on next page*

*Figure 5 continued*

DOI: https://doi.org/10.7554/eLife.27995.015

The following source data and figure supplement are available for figure 5:

**Source data 1.** Post-stimulus responses of replay cells before and after Wulst lesion in both hemispheres.
DOI: https://doi.org/10.7554/eLife.27995.017
**Figure supplement 1.** Locations of bilateral Wulst lesions in the pigeon brain.
DOI: https://doi.org/10.7554/eLife.27995.016

(one-way ANOVA, $F_{5, 168} = 4.03$, $p = 0.002$, $n = 29$ cells; *Figure 5E*) and the peak activities of the replay cycles decreased over time ($F_{5, 168} = 9.96$, $p < 0.0001$; *Figure 5F*). Therefore, these data from the Wulst-lesioned animals further confirmed that the pallial Wulst did not participate in the modulation of the post-stimulus oscillatory response of the DLA replay cells.

# Discussion

The present study showed that thalamic cells in the pigeon DLA, which encoded ambient luminance, could be entrained by repetitive LD stimuli, and retained the entrained oscillatory activities even after the periodic stimulation was terminated. Moreover, the post-stimulus replay responses of these cells depended on the temporal frequency and the number of LD cycles applied during LD stimulation. The feedback inputs of the pallial Wulst to the DLA did not contribute to the post-stimulus oscillatory activities of the DLA cells. Our results suggest that thalamic cells not only follow the current changes of periodic stimuli, but also remember the temporal patterns of periodic stimuli experienced previously.

## Thalamic cells more reliably follow the temporal patterns of repetitive stimuli than retinal cells

Previous study has shown that retinal ganglion cells in salamander and mouse retinae respond to an omitted stimulus in a sequence of flashes (*Schwartz et al., 2007*). However, the retinal response to the omitted stimulus is not observed more than once and it is irrelevant to whether the recorded cells respond to the flashes throughout the flash sequence. Compared to the retinal cells, the pigeon thalamic cells in our experiment not only accurately signaled the time of each luminance change of repetitive stimuli during LD stimulation, but also showed reliable replay responses after the periodic stimulation. The omitted stimulus response in the retina occurs for repetitive stimuli with short intervals (20 ms–100 ms) (*Schwartz et al., 2007*), but thalamic cells can be entrained by temporal patterns with long intervals (10 s–8 min). Furthermore, the replay responses of thalamic cells can be maintained for more than one cycle. For example, the replay responses of the DLA cell in *Figure 3* continued for up to 48 min and 16 replay cycles in constant darkness after 20 min of periodic LD stimulation (L/D: 60 s/60 s, 10 cycles). In addition, the number of replay cycles increased with the increase in the number of LD cycles.

The post-stimulus oscillatory responses of thalamic replay cells entrained by periodic stimuli with the interval in the seconds-to-minutes range are unlikely to originate from the retina. The omitted stimulus potential (OSP) is traditionally regarded as a sign of expectation of a stimulus at the due-time. In human EEG recordings, the OSP has been observed under low (< 2 Hz) and fast (> 5 Hz) stimulus rates (*Bullock al., 1994*). The fast OSP arises in the retina (*Bullock al., 1990*; *Schwartz et al., 2007*), but it is not clear whether the slow OSP also arises in the retina. When a periodic stimulus with an interval in the order of seconds was presented to zebrafish larvae in vivo, the retinal ganglion cells did not show post-stimulus rhythmic activity (*Sumbre et al., 2008*). In addition, during visual conditioning, the responses of retinal ganglion cells evoked by CS (a few seconds of whole-field light) did not change with the training of pigeons (*Wild and Cohen, 1985*). These studies imply that retinal ganglion cells might not retain the entrained activity after the presentation of repetitive stimuli with the interval in the seconds-to-minutes range.

## Experience-dependent representation of time interval is not confined to the pallial/cortical level in the brain

In addition to the thalamic cells reported in the present study, the post-stimulus replay response has also been observed in the optic tectum of zebrafish larvae (*Sumbre et al., 2008*). There are two major visual pathways linking the eyes to the brain: one projects to the visual thalamus and the other to the optic tectum of vertebrates or the superior colliculus of mammals (*Grüsser et al., 1975*; *Jessell et al., 2000*). In visual conditioning, after the repetitive CS presentation of seconds in duration, neuronal ensembles in the zebrafish tectum show rhythmic activities with an interval matching the duration of the CS. Correspondingly, the visuomotor behavior of zebrafish larvae is highly correlated with the post-CS rhythmic neuronal activities in the tectum (*Sumbre et al., 2008*). Therefore, the experience-dependent representation of time interval might not be confined to the pallial/cortical level, but may occur as early as the subcortical levels in the brain.

Although repetitive visual stimuli induce similar replay responses in the tectum and thalamus, the temporal information encoded in these two nuclei may contribute to different time-dependent tasks. Through descending outputs to the hindbrain, the vertebrate optic tectum/mammalian superior colliculus can use detected temporal information to accurately control the fast and immediate movements of animals, such as the eye-head coordination of monkeys (*Klier et al., 2003*), the prey capture and visuomotor behavior of zebrafish (*Gahtan et al., 2005*; *Sumbre et al., 2008*), and the looming-object detection and avoidance of pigeons, cats, and mice (*Wu et al., 2005*; *Liu et al., 2011*; *Shang et al., 2015*). However, the avian DLA/mammalian LGN mainly project to the pallium/cortex, thus it is more likely that the visual thalamus participates in the perceptual and cognitive tasks performed by the pallium/cortex. In the traditional view, the thalamus is thought to passively transfer ongoing visual information from the retina to the cortex (*Derrington, 2001*; *Liu et al., 2008*; *Naito et al., 2013*). By contrast, the thalamic replay cells in the present study not only followed the changes of current stimuli, but also retained a copy of the periodic events exposed previously. Here, these cells acted like a time-adjustable alarm clock. Although the external periodic events were vanished, the replay cells continuously signaled the time of upcoming events that would occur. Given the recent findings that the visual thalamus participates in many dynamic processes in the visual pathway (*Cudeiro and Sillito, 2006*; *Guillery and Sherman, 2011*; *Saalmann and Kastner, 2011*; *Sherman, 2016*), the timing signal that we observed in the visual thalamus might be recruited by the pallium/cortex in the time-dependent task.

Given that neither pharmacological inactivation nor electrolytic lesions of the Wulst in both hemispheres affected the post-stimulus oscillatory activities of the DLA cells entrained by external, slow frequency LD periodic stimulation, the time-interval representation of avian thalamic cells in the order of minutes is unlikely to be modulated by the pallium Wulst. Like the mammalian LGN, the avian DLA is involved in far more than the simple transmission of visual information from the retina to the visual pallium. In addition to the retinal and pallial Wulst projections, the DLA also receives afferent supplies from the suprachiasmatic nucleus (SCN) and the optic tectum (*Miceli et al., 2008*; *Cantwell and Cassone, 2006*). However, we do not know which cognitive functions of visual thalamus are modulated by the inputs from SCN and tectum in the pigeon. Therefore, further neurophysiological evidence is required to reveal the possible modulating inputs to DLA cells that are needed for time-interval representation.

## Post-stimulus oscillatory responses of the thalamic replay cells are induced by an intrinsic circuit in the thalamus

The intrinsic electrical properties of thalamic cells might determine their oscillatory responses to periodic stimuli. In mammals, thalamocortical cells (TC cells) are excitatory and project to the cortex, whereas the local interneurons in the LGN are GABAergic and exert inhibitory influence on TC cells (*Uhlrich and Cucchiaro, 1992*; *Sherman, 2016*). Previous studies on thalamic slices suggest that TC cells and interneurons exhibit voltage-dependent intrinsic oscillation (*Zhu et al., 1999*; *Llinás and Steriade, 2006*). In guinea pig thalamic slices, TC cells show voltage-sensitive ionic conductance and can generate two distinct functional states: repetitive spiking and bursting modes (*Llinás and Jahnsen, 1982*). By adjusting the membrane potential, the firings of TC cells can be switched from one state to the other. The interplay between low-threshold $Ca^{2+}$ ($I_{Ca}$) and $Na^{+}$-$K^{+}$ current ($I_{Na+K}$) is crucial for the low-frequency oscillation (< 4 Hz) of TC cells (*McCormick and Pape, 1990*;

*Soltesz et al., 1991*). In comparison with TC cells, the interaction between $I_{Ca}$ and the calcium-activated non-selective cation current ($I_{CAN}$) is essential for the oscillatory burst firing of interneurons (*Bal and McCormick, 1993*; *Zhu et al., 1999*).

Our experiment provides further evidence that the intrinsic circuit in the DLA, rather than the feedback inputs from the pallial Wulst, plays the primary role in the post-stimulus replay responses of DLA cells. The avian DLA receives direct retinal inputs and has reciprocal connections with the pallial Wulst. The retinal inputs are unlikely to contribute to the entrained post-stimulus activity in the seconds-to-minutes range, as discussed above. Neither pharmacological inactivation nor electrolytic lesions of the Wulst in both hemispheres had any significant effect on the post-stimulus oscillatory activities of DLA replay cells.

To understand the potential mechanism underlying the post-stimulus replay responses of the DLA cells, we introduced a simplified computational model (see Materials and methods) based on the electrophysiological properties of thalamic cells reported in previous studies (*Zhu et al., 1999*; *Llinás and Steriade, 2006*) and our current results. The simple two-cell system included two model neurons (R-neuron and NR-neuron) that respectively simulated a replay cell and a non-replay cell. The DLA in pigeons has an abundance of $GABA_A$, $GABA_B$, and benzodiazepine-binding sites (*Veenman et al., 1994*). In the model, we proposed that the R-neuron received inhibitory synaptic inputs from the NR-neuron ($I_{syn}$), but the NR-neuron did not receive synaptic inputs from the R-neuron (*Figure 6A*). By adjusting the model parameters, the model neurons captured most of the response features of the thalamic cells observed in the present experiment (*Figure 6B*). During LD stimulation, both the R-neuron and NR-neuron exhibited oscillatory activities synchronous with the periodic LD stimuli. After LD stimulation, the R-neuron continued the entrained oscillatory activity, whereas the NR-neuron returned to the non-oscillatory firing pattern. Further analyses of the post-stimulus activities of the R-neuron (R-neuron [u, $I_{syn}$]) showed that the model neuron had a response pattern consistent with that of the real cell illustrated in *Figure 3*, as reflected in the increasing duration and decreasing mean activity of each replay cycle over time (*Figure 6C,D*). Without inhibitory inputs ($I_{syn}$), the model neuron (R-neuron [u]) still showed oscillatory activity after LD stimulation, but the duration and mean activity of each replay cycle did not change over time (*Figure 6C,D*).

The computational model is not fully conclusive and only provides a possible explanation for the neural mechanism underlying the declining oscillatory responses of thalamic cells after periodic stimulation. In addition to the intrinsic membrane currents applied in the present model (*Marder et al., 1996*), there are several factors that might affect the persistent responses of thalamic cells in the absence of periodic inputs, such as N-methyl-D-aspartate (NMDA) currents (*Wang, 2001*; *Bottjer, 2005*) and short-term synaptic plasticity in the neural network (*Mongillo et al., 2008*; *Szatmáry and Izhikevich, 2010*). In addition, interneurons can exert inhibitory influences on neurons projecting to the Wulst in the avian DLA (*Miceli et al., 2008*). It is plausible that the inhibitory inputs from the NR-neuron to the R-neuron in the model were mediated by inhibitory interneurons. In the present study, the computational model was simplified to include only two cells (R-neuron and NR-neuron). In the real brain, however, the post-stimulus oscillatory responses of replay cells are very likely to be generated by neural networks composed of a population of neurons in the DLA rather than a single R-neuron (*Gutnisky and Dragoi, 2008*; *Wang et al., 2011*; *Benucci et al., 2013*).

## Replay responses of thalamic cells are not induced by anesthesia

As slow wave activity (< 1 Hz) emerges in the thalamic cells of both anesthetized (*Steriade et al., 1993*) and awake animals (*Albrecht et al., 1998*; *Filippov and Frolov, 2005*), one could assume that the anesthetic state induced the oscillatory activities of thalamic cells observed in the present experiment. However, our results suggest that the post-stimulus replay response of the thalamic cells was evoked by the external stimuli rather than by the anesthetic state for six reasons: (1) both replay and non-replay cells exhibited steady spontaneous activities under constant photic conditions before periodic stimulation; (2) both replay and non-replay cells showed synchronous activities with luminance changes during LD stimulation; and (3) under constant photic conditions after periodic stimulation, replay cells continued the entrained oscillatory activities in contrast to non-replay cells. The post-stimulus replay responses of replay cells were maintained for a long period of time, and then gradually returned to the non-oscillatory responses over time. (4) The replay responses of

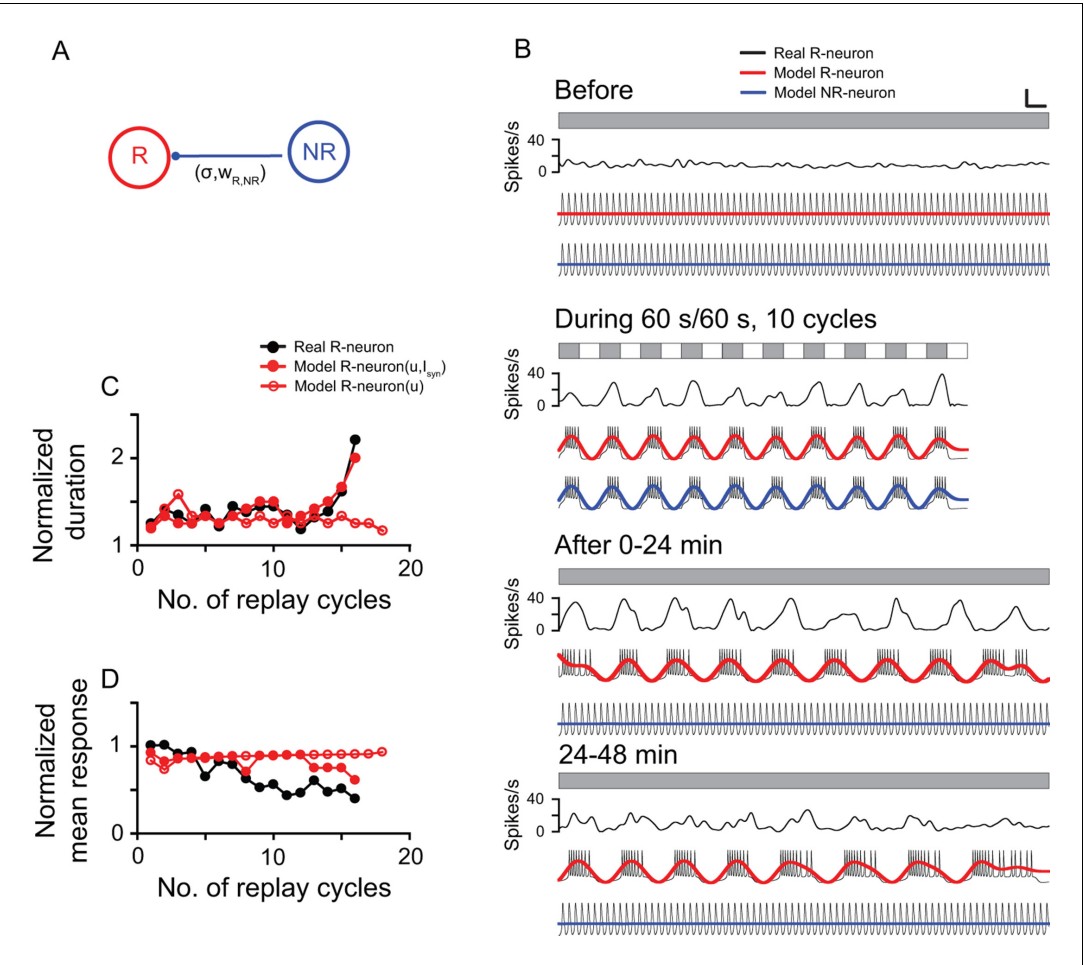

**Figure 6.** Computational model simulation of the responses of thalamic luminance cells to periodic stimuli. (A) In the simplified neural network, the model replay neuron (R-neuron) received inhibitory synaptic inputs ($I_{syn}$) from the non-replay neuron (NR-neuron) with synaptic weight ($w_{R,NR}$) and coupling strength ($\sigma_R$). The model NR-neuron did not receive synaptic inputs from the R-neuron. (B) By adjusting the simulation parameter values (see Materials and methods), the model R-neuron (red line) and NR-neuron (blue line) captured most of the response features of the thalamic cell in *Figure 3* (black line) to the periodic LD stimulus (LD: 60 s/60 s, 10 cycles). For clear illustration of the response changes, the simulated firing traces of each model neuron were filtered by a zero-phase, low-pass Butterworth filter. The filtered firing rates are marked in red (R-neuron) and blue (NR-neuron). (C, D) Like the real replay cell illustrated in *Figure 3*, after LD stimulation, the duration of each replay cycle of the R-neuron (R-neuron [u,$I_{syn}$]) increased (C), while the mean activity of each replay cycle decreased with time (D). Without the inhibitory inputs of the NR-neuron ($I_{syn}$), the R-neuron (R-neuron [u]) showed oscillatory activity, but the duration and activity of each replay cycle did not change over time. u: recovery variable in the equation of the model neuron. Scale bars: 1 min, 300 µV in (B) for the firing traces of the model neurons. For other conventions, see *Figure 1*
DOI: https://doi.org/10.7554/eLife.27995.018

The following source data is available for figure 6:

**Source data 1.** Simulation results of model neurons to periodic photic stimuli.
DOI: https://doi.org/10.7554/eLife.27995.019

thalamic cells after LD stimulation were regulated by the temporal frequency and number of LD cycles applied during LD stimulation. (5) The time interval of LD stimuli shorter than 10 s (L/D: 5 s/5 s) could not induce the post-stimulus oscillatory responses of replay cells. (6) During recording, the depth of anesthesia was monitored and additional top-up doses of anesthetic were applied as required. Moreover, animals were also isolated from any other sensory stimuli, such as auditory, olfactory, and taste stimuli in the environment. Taken together, the periodic stimulus was the only

factor that induced the photic and replay responses of the thalamic cells. In addition, previous studies have reported that the abilities of thalamic cells to discriminate visual features are less affected by the brain state (alert/non-alert) of animals. Although thalamic cells have higher firing rates when animals are awake than during anesthetic state, their sensitivities to stimuli with different spatial and temporal contrasts are comparable under the two conditions (*Cano et al., 2006*; *Alitto et al., 2011*).

## Functional significance of the entrained time-interval representation in the avian thalamus

The present study sheds light on the neural mechanism of translating temporal information from external and variable photic events into internal timing in the brain. Our study provides the first evidence that the visual thalamus is involved in time perception and in memorizing the timing of periodic events that have occurred previously. The retina-DLA-Wulst pathway in birds participates in spatial and sun compass orientation as well as in light-dependent navigation (*Budzynski et al., 2002*; *Heyers et al., 2007*; *Watanabe et al., 2011*; *Keary and Bischof, 2012*; *Bischof et al., 2016*). The entrained replay responses of thalamic cells may contribute to the navigation behaviors of birds by signaling the time of expected events previously experienced in the environment.

# Materials and methods

## Animal preparation

All experiments performed on the 38 adult homing pigeons (*Columba livia*) were in accordance with the guidelines for the care and use of animals established by the Society for Neuroscience and approved by the Institutional Animal Care and Usage Committee (IACUC) of the Institutes of Biophysics, Chinese Academy of Sciences (SYDK2016-07).

Each pigeon was initially anesthetized by injecting ketamine hydrochloride (40 mg/kg) and xylazine hydrochloride (5 mg/kg) into the pectoral muscles, and was supplemented with ketamine hydrochloride (20 mg/kg) and xylazine hydrochloride (2 mg/kg) per hour. The animal was gently wrapped in a bag and placed on a foam-lined holder in a stereotaxic apparatus. The depth of anesthesia was monitored by breathing patterns and reflex from pinching the toe. Body temperature was maintained at 41°C by a warming pad. The wound edges and muscles were infiltrated periodically with lidocaine.

The telencephalon overlying the DLA (anterior [A]: 6.25 to 7.25, lateral [L]: 2.80 to 3.80, height [H]: 7.00 to 8.00) and the Wulst (A 8.00 to 14.50, 0 to L3.00, 0 to H3.00) in both hemispheres was exposed with a dental drill and surgical forceps (*Karten and Hodos, 1967*). To investigate the possible contributions of the inputs from the pallial Wulst to the post-stimulus oscillatory activities of the DLA cells, the Wulst in both hemispheres was pharmacologically inactivated or electrolytically lesioned. (1) Pharmacological inactivation was achieved via the administration of a 1 μl Hamilton syringe filled with 2% muscimol (Abcam, UK). Multi-unit recordings show that 1 μl of muscimol (2%) can completely inactivate neuronal activities in a 2 mm diameter area around the cannula tip for ~3 hr, and can significantly attenuate the neuronal activities in a 4 mm diameter area (*Partsalis et al., 1995*; *Arikan et al., 2002*). To inactivate the Wulst in both hemispheres, we injected 1 μl of muscimol (2%) at four different coordinates: A10, L1.5, H1.5 and A12, L1.5, H1.5 in the left hemisphere, and A10, L1.5, H1.5 and A12, L1.5, H1.5 in the right hemisphere. The locations of injection sites were confirmed by injecting 0.2 μl of direct-blue 15 (2% direct-blue 15 in 0.5 M sodium acetate solution, Sigma, USA). (2) Electrolytic lesions (CH-HI Cautery, Advanced Meditech International, USA) were applied to the Wulst in both hemispheres. During surgery, the physical condition of the anesthetized animals was strictly monitored, with indicators including breathing and heart rate. After surgery, the lesion area was covered with sterilized medical hemostatic sponge. The electrophysiological recordings started 1 hr after surgery.

## Photic stimulation

One eye of each animal was occluded, and the other eye was stimulated by a light-emitting diode (LED). The center of the LED was in line with the optical axis of the viewing eye and 1.5 cm from the eyeball. The light from the LED was diffused over the whole visual field of the viewing eye. A rubber

eye cap enclosing the LED was fitted closely to the eye orbit rim. The eye-cap and LED light constituted the probe of a custom-designed multifunctional visual photostimulator (Institute of Biophysics, Chinese Academy of Sciences, Beijing, China).

For the photic stimulus (423 nm–688 nm), the luminance level was adjusted in six steps (0, 20, 50,100, 200, and 400 lux). The duration of each LD cycle (2, 4, 10, 20, 30, 60, 120, 240, and 480 s) and number of LD cycles (5–25 cycles) were also adjusted. In the routine experiments, the dark was 0 lux. The luminance of the photic stimulus was measured by a digital light meter (LX-1330B, Shenzhen TONDAJ Instrument Co., China).

## Recordings

Single-unit recordings were made in the pigeon DLA/Wulst using tungsten-in-glass microelectrodes made in the laboratory (2–3 MΩ). Luminance cells in the DLA were first isolated with a flashlight. Two rigorous criteria were used to identify luminance cells: (1) in constant light or darkness, the steady firing of the recorded cell was sustained over a long period of time (usually 20–60 min) before the repetitive LD stimulus was presented; and (2) the firing rates of the recorded cell either increased or decreased monotonically when the luminance level was increased from 0 to 400 lux in 3–6 steps. The firing rates for each luminance level were averaged for 5–10 repeats.

The activities of each luminance cell were examined before, during, and after 5–25 LD cycles of photic LD stimulation. After LD stimulation, the responses of each cell were continuously recorded for 2–3 hr. Light-activated cells were tested under light, and light-suppressed cells were tested in darkness until the cells completely returned to spontaneous activity. When the photostimulator sent a switch signal to the LED, it also sent the signal simultaneously to the computer to mark the time of stimulus onset/offset. Neuronal spikes were amplified and fed into an oscilloscope (54622A, Agilent Technologies Inc., USA) for observation and a computer for data collection and off-line analyses.

Recording sites were identified via electrolytic lesions that were generated by applying positive currents of 30–100 μA for 20–30 s. To verify the electrolytic lesion sites and direct-blue marks in the brain, the animals were euthanized by an overdose of urethane (4 g/kg) via intraperitoneal injection after the experiment. The brains removed from the skulls were fixed in 4% paraformaldehyde for 6–12 hr, and soaked in 30% sucrose solution in a refrigerator (4°C) overnight. Frontal sections were cut on a freezing microtome (Leica CM1850, Germany) at 40 μm thickness and counterstained with cresyl violet (Sigma, USA). They were dehydrated and covered for subsequent microscopic observations.

## Data analyses

Neuronal spikes and photostimulator switch signals were sampled at 8000 Hz with Cool Edit software (Version 2.0, Syntrillium Software Co., USA). The data were quantitatively analyzed off-line by Spike2 software (CED, Cambridge Electronic Design Ltd., UK) and custom-made MATLAB routines (R2009a, MathWorks, USA). Single units were classified on the basis of full wave templates and clustered by principle component analysis and direct waveform feature measures. Only well-isolated units were included in this study.

To determine whether there was a significant oscillation in each response histogram (bin = 0.2 s), we constructed a control group composed of 1000 resampled histograms computed by bootstrap re-sampling of the original response histogram. The original response and reconstructed histograms in the control group were transformed into the frequency domain by Fast Fourier Transformation (FFT) using Hanning windowing. The oscillation frequency of the original histogram detected by FFT analysis was considered statistically significant only when its FFT amplitude was larger than the mean +4 SDs of the control.

To separate each replay cycle of the recorded cell precisely, the response histogram was filtered by a zero-phase, low-pass Butterworth filter. The cutoff frequency of the filter was the maximal oscillation frequency detected by shifting an analysis window (7 min) at 1 min steps from the end of the periodic LD stimulation to the end of the recording. The start and end time points of each replay cycle was the intersection of the filtered histogram with the mean value of the filtered histogram itself.

To measure the correlation between the responses of each cell and the luminance level of the photic stimulus, the mean response of the recorded cell to each luminance level (5–10 repeats) was

normalized by the cell's maximal response to all tested luminance levels. To quantify the changes in oscillatory activities of the real or model cells after LD stimulation, the duration of each replay cycle of the real and model cells was normalized by the duration of the LD cycle. Correspondingly, the mean or peak response of the replay cycle of the light-activated/light-suppressed cells was normalized by the mean response averaged across all light/dark periods during LD stimulation, respectively.

## Computational model

The model neuron (*Morris and Lecar, 1981*; *Sherman and Rinzel, 1992*; *Izhikevich, 2007*) was described by the following equations:

$$C\dot{V} = I + I_1 + I_K + I_{Na} + I_{Ca}$$

$$I_1 = g_1(V_1 - V)$$

$$I_k = g_k s(V_k - V)$$

$$I_{Na} = g_{Na} m_\infty(V)(V_{Na} - V)$$

$$I_{Ca} = g_{Ca} u(V_{Ca} - V)$$

$$\dot{s} = \lambda(V)(w_\infty(V) - s)$$

$$w_\infty(V) = \frac{1}{2}\left(1 + \tanh\frac{V - V_1}{V_2}\right)$$

$$m_\infty(V) = \frac{1}{2}\left(1 + \tanh\frac{V - V_3}{V_4}\right)$$

$$\lambda(V) = \frac{1}{3}\cosh\frac{V - V_1}{2V_2}$$

where V (mV) and C (μF) are the membrane potential and capacitance of the model neuron, respectively; $I$, $I_l$, $I_k$, $I_{Na}$, and $I_{Ca}$ are the applied current and the currents of leak, K$^+$, Na$^+$, and Ca$^{2+}$ respectively; $g_l$, $g_k$, $g_{Na}$, and $g_{Ca}$ are the leak, K$^+$, Na$^+$, and Ca$^{2+}$ conductances through membrane channels, respectively; $V_l$, $V_k$, $V_{Na}$, and $V_{Ca}$ are the equilibrium potentials of the relevant leak, K$^+$, Na$^+$, and Ca$^{2+}$ channels, respectively; $V_2$ and $V_4$ are the reciprocal of the slope of voltage dependence of $w_\infty(V)$ and $m_\infty(V)$, and $V_1$ and $V_3$ are the potentials whose values are dependent on $V_2$ and $V_4$, respectively. By adjusting the dynamic variable 'u', the model neuron shows continuous spiking or bursting. A constant u value induces the repetitive spiking of the model neuron. When the changes in the u value satisfy the following equation:

$$\dot{u} = \mu_{R/NR}(w_\infty(V) - u)$$

the model neuron shows bursting, where $\mu_{R/NR}$ determines the inter-burst interval of the R-neuron or NR-neuron, respectively. The u value slowly increases when the neuron is depolarized and slowly decreases when the neuron is hyperpolarized, which periodically switches the model neuron between the active and silent phases (*Sherman et al., 1988*).

Using the model neuron described above, we set up a simple two-cell system composed of two model neurons (R-neuron and NR-neuron) that simulated a replay cell and a non-replay cell, respectively (*Figure 6A*). We proposed that the model R-neuron received inhibitory synaptic inputs from the model NR-neuron, with synaptic weight ($w_{R,NR}$) and coupling strength ($\sigma_R$), as described by the following equations:

$$C\dot{V}_R = I + I_1 + I_k + I_{Na} + I_{Ca} + I_{syn}$$

$$I_{syn} = \sigma_R w_{R,NR}(E_{syn} - V_R)$$

$$\dot{w}_{R,NR} = \arctan[V_R(V_{NR} - V_R w_{R,NR})]$$

The model NR-neuron did not receive synaptic inputs from the model R-neuron, and its neuronal activity can be described by the following equation:

$$C\dot{V}_{NR} = I + I_1 + I_K + I_{Na} + I_{Ca}$$

where $V_R$ or $V_{NR}$ denotes the membrane potential of the model R-neuron or NR-neuron, respectively. The $E_{syn}$ value depends on whether the synapse is excitatory or inhibitory.

To simulate the real thalamic cells observed in the present experiment, the parameters in the equations were adjusted: C = 3.33 μF, V = -0.041 mV, s = 0.5048, $I_{light/dark}$ = 0.06 μA/0.1 μA, $g_{Ca}$ = 1 mS, $V_{Ca}$ = −0.7 mV, V = −0.041 mV, $V_1$ = 0.1 mV, $V_2$ = 0.05 mV, $V_3$ = −0.01 mV, $V_4$ = 0.15 mV, $g_l$ = 0.5 mS, $V_l$= −0.5 mV; $g_k$ = 2 mS, $V_k$ = −0.7 mV, $g_{Na}$ = 1.2 mS, $V_{Na}$ = 1 mV, $E_{syn}$ = −0.7 mV, $w_{R,NR}$ = 0.05. For the R-neuron, $\sigma_R$ = 0.036, u = 0.075 before LD stimulation, $\sigma_R$ = 0.086, $\mu_R$ = 0.01 during LD stimulation. The $\sigma_R$ and $\mu_R$ values changed with time (t): $\sigma_R$ (t) = 0.0091–0.0005$e^{0.00008t}$, $\mu_R$ (t) = 0.008+0.001log(0.005 t)/log0.05 after LD stimulation. For the NR-neuron, u = 0.075 before and after LD stimulation, and $\mu_{NR}$ = 0.01 during LD stimulation.

## Acknowledgements

We thank Drs Bing Li, Da-Peng Li, and Peng Cao for discussion on data analyses and Xiu-Chun Wang, Bo-Hai Lin, Su Liu, Xu-Dong Zhao, Xiao-Fei Guo, and Rui-Min Zheng for technical assistance. We also thank Drs Felix Stroeckens, Maik C Stüttgen, Zhi-Hua Wu, and Chuan Zhang for reading this manuscript and for scientific discussions. The National Institute of Metrology (China) measured the spectrum and optical power of the LED light. This work was supported by grants from the National Natural Science Foundation of China to QX (31372206) and YW (31371105) , and grants from Chinese Academy of Sciences to YY (QYZDB-SSW-SMC032) .

## Additional information

### Funding

| Funder | Grant reference number | Author |
|--------|------------------------|--------|
| National Natural Science Foundation of China | 31372206 | Qian Xiao |
| National Natural Science Foundation of China | 31371105 | Yi Wang |
| Chinese Academy of Sciences | QYZDB-SSW-SMC032 | Yan Yang |

The funders had no role in study design, data collection and interpretation, or the decision to submit the work for publication.

### Author contributions

Yan Yang, Conceptualization, Formal analysis, Investigation, Methodology; Qian Wang, Formal analysis, Investigation, Methodology; Shu-Rong Wang, Conceptualization, Visualization, Methodology, Writing—original draft; Yi Wang, Conceptualization, Funding acquisition, Visualization, Writing—original draft, Project administration, Writing—review and editing; Qian Xiao, Conceptualization, Software, Formal analysis, Funding acquisition, Investigation, Visualization, Methodology, Writing—original draft, Project administration, Writing—review and editing

## Author ORCIDs

Yan Yang https://orcid.org/0000-0003-0535-9824
Yi Wang http://orcid.org/0000-0002-4448-1809
Qian Xiao http://orcid.org/0000-0002-5946-8607

## Ethics

Animal experimentation: This study was performed in strict accordance with the recommendations in the guidelines for the care and use of animals established by the Society for Neuroscience and approved by the Institutional Animal Care and Usage Committee (IACUC) of Institutes of Biophysics, Chinese Academy of Sciences (SYDK2016-07).

## Decision letter and Author response

Decision letter https://doi.org/10.7554/eLife.27995.022
Author response https://doi.org/10.7554/eLife.27995.023

## Additional files

### Supplementary files

• Transparent reporting form
DOI: https://doi.org/10.7554/eLife.27995.020

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
