## [Decision Letter]

Thank you for submitting your article "Representation of time interval entrained by periodic stimuli in the visual thalamus of pigeons" for consideration by *eLife*. Your article has been reviewed by two peer reviewers, and the evaluation has been overseen by a Reviewing Editor and Timothy Behrens as the Senior Editor. One of the reviewers, Gregory William Schwartz, has agreed to share his name.

The reviewers have discussed the reviews with one another and the Reviewing Editor has drafted this decision to help you prepare a revised submission.

This manuscript reports a very unexpected, long-lasting resonance in the visual thalamus of pigeons. Learning and production of time sequences is an important task that allows animals to plan and perform complex tasks where time keeping is required. While a fair amount of work has been performed at the level of cortical sensory areas, basal ganglia and cerebellum, the authors investigate the possibility that periodic sequences can also be learned at the level of dorsolateralis anterior thalami (DLA), the avian equivalent to the mammalian LGN. The authors uncover a subset of neurons in DLA that, once primed by slow (<<1Hz) periodic changes in full field illumination, persist in expressing oscillatory activity for up to one hour. The main results from electrophysiological experiments are remarkable and certainly deserve attention. Electrophysiological experiments seem to be conducted carefully, raw recordings show that they can detect spikes with high SNR ratio and downstream analyses are fairly well described.

Summary:

1) With regard to the frequency range used for the experiments, Intervals from 30 s to 240 s are explored with no strong rationale for that choice. The authors should perform additional experiments to explore a broader range to determine where this phenomenon breaks down at both the high and low frequency ends.

2) The role of thalamocortical interactions in generating slow oscillatory activity has been subject to a large body of work. As a cortical feedback is excluded in authors' model they have to provide some experimental evidence for the possibility that DLA alone is capable of sustaining periodic activity without feedback from visual Wulst. A way to do that would be by pharmacological inhibition of the visual Wulst. The authors should perform additional experiments to explore this issue.

3) The model is too speculative. It either should be removed from the main body of the paper (and perhaps incorporated into the Discussion) or evidence should be provided to support its components. Specifically: a) The oscillatory behavior is all captured in the "u" variable that has no clear connection to a biophysical property of calcium channels. Without additional measurements, this parameter ends up being simply a recapitulation of the phenomenon. Measurements of the particular conductances that cause the resonance (if that is even the source – it could be a network property) are beyond the scope of this study. b) While inhibition from non-replay cells is one possible mechanism for the decay back to baseline, there is no experimental evidence for this, and (as the authors acknowledge in the Discussion), there are plenty of alternatives. Overall, the model seems like reasonable speculation for a Discussion section, but not a real result. c) The asymmetric connection between R and NR neurons in the model is somewhat arbitrary. The authors should discuss anatomical justification of this connectivity based on known anatomical data.

[Editors' note: further revisions were requested prior to acceptance, as described below.]

Thank you for resubmitting your work entitled "Representation of time interval entrained by periodic stimuli in the visual thalamus of pigeons" for further consideration at *eLife*. Your revised article has been favorably evaluated by Timothy Behrens (Senior editor), a Reviewing editor, and two reviewers.

The manuscript has been improved but there are some remaining issues that need to be addressed before acceptance, as described by reviewer #2. We envision that these revisions will be straightforward to carry out.

*Reviewer #1:*

In my view, the authors have done an excellent job addressing the reviews with additional experiments to test the robustness of the effect and whether it depends on feedback from Wulst. Moving the model to the Discussion helps to identify it as more a theory to inspire future experiments than an actual well-constrained result.

*Reviewer #2:*

Overall the authors addressed my concerns. However there are still a couple of more minor issues to consider:

The authors performed two series of experiments to investigate the possibility that the emergence of replay cells could be due to a feedback from the visual Wulst.

· Pharmacological inactivation. This is thoroughly described in Materials and methods and in Figure 4—figure supplement 1. However they lack a positive control – an electrophysiological recording in HA showing that they can effectively shut down neuronal activity in this area. If this were available it would increase the value of these data.

· Surgical lesion. The procedure, animal recovery and lesioning parameters (e.g. current, duration) are not well described in the Materials and methods section, although Figure 5—figure supplement 1 clearly shows that their procedure is very effective. A bit more description in the Materials and methods would be helpful.

The finding that both procedures do not have any effect on the ability of some DLA cells to replay periodic luminance stimuli is convincing. However, given the extent of these procedures, especially the electrolytic lesioning, I would expect to see some nonspecific effects such as a global reduction/increase in basal or evoked firing rates. As firing rates are normalized in order to compare peak responses (as in Figure 4, Figure 5) these nonspecific effects cannot be inferred from the data presented. A panel with raw firing rates, like Figure 1 before and after pharmacological block and/or electrolytic lesions would be helpful in building the argument that periodic replay is selectively resistant to these modifications.

In case no nonspecific effects can be observed the authors should provide some additional lines in Discussion explaining that outcome by providing plausible reasons as to why eliminating feedback from the visual Wulst had no impact on DLA activity in their recordings.

---

## [Author Response]

Summary:1) With regard to the frequency range used for the experiments, Intervals from 30 s to 240 s are explored with no strong rationale for that choice. The authors should perform additional experiments to explore a broader range to determine where this phenomenon breaks down at both the high and low frequency ends.

To address this issue, we performed a new experiment to explore the frequency threshold of periodic stimulus that can induce the oscillatory activities of replay cells after LD stimulation. The shortest interval of periodic LD stimulus was set to 2 s (L/D: 1 s/1 s, 0.5 Hz). The longest interval of periodic stimuli was set to 480 s (L/D: 240 s/240 s, 0.002 Hz), which was the lowest frequency that we could test under current experimental conditions. Regardless of the recording time before LD stimulation, it took 1 hour and 20 minutes to present 10 LD cycles with duration in 480 s (L/D: 240 s/240 s). Neuronal activities of each luminance cell after LD stimulation were normally continuously recorded for up to 2-3 hours. Therefore, when the stimulus interval was longer than 480 s, it is hard to maintain a cell for a long enough time to collect all experimental results before, during, and after LD stimulation. Therefore, the duration of LD cycle of applied periodic photic stimuli included 1 s/1 s, 2 s/2 s, 5 s/5 s, 10 s/10 s, 15 s/15 s, 30 s/30 s, 60 s/60 s, 120 s/120 s and 240 s/240 s.

We found that 1) As shown in Figure 1, the luminance cells in the present experiment encoded the ambient luminance. The DLA replay cells did not show oscillatory responses after LD stimulation when the interval of periodic stimulus applied during LD stimulation was shorter than 10s (L/D: 5 s/5 s). 2) As shown in Figure 2, the post-stimulus oscillatory activities of replay cells fitted better with applied LD stimuli when the temporal frequency of LD stimuli was in the range 0.0167 Hz (L/D: 30 s/30 s) to 0.0042 Hz (L/D: 120 s/120 s). Although we tested LD stimuli with periods up to only 480 s, it is conceivable that these neurons could learn to replay the much longer LD cycle and establish a stable rhythmic firing after entrained with enough number of LD cycles.

2) The role of thalamocortical interactions in generating slow oscillatory activity has been subject to a large body of work. As a cortical feedback is excluded in authors' model they have to provide some experimental evidence for the possibility that DLA alone is capable of sustaining periodic activity without feedback from visual Wulst. A way to do that would be by pharmacological inhibition of the visual Wulst. The authors should perform additional experiments to explore this issue.

As the thalamocortical inter-connections in the mammalian brain, the avian DLA has reciprocal connections with the pallial Wulst (equivalent to the mammalian cortex) in both hemispheres (Karten et al., 1973; Miceli et al., 1987; Miceli et al., 2008). To address the question whether the feedback inputs from Wulst to DLA contribute to the oscillatory activities observed in the DLA, we designed two experiments to compare the activities of DLA cells before and after the Wulst in both hemispheres was pharmacologically inactivated or damaged by electrolytic lesion, respectively.

Two types of luminance cells were recorded in the present experiment: replay cells and non-replay cells. Driven by the retinal inputs, both replay and non-replay cells accurately encoded the luminance changes of periodic stimuli during LD stimulation. However, under constant photic conditions after LD stimulation, only replay cells retained the entrained oscillatory activities. Therefore, in the additional experiments, we mainly concerned about whether the response patterns of replay cells after periodic stimulation were affected by pharmacological inactivation or electrolytic lesion of the Wulst in both hemispheres.

Experiment one:

The previous studies have carefully measured the effective spread of 1 µl of muscimol (2%) in the brain (Partsalis et al., 1995; Arikan et al., 2002). It is known that 1 µl of muscimol (2%) can completely inactivate neuronal activities within 2 mm in diameter around the cannula tip at roughly 3 hr and significantly attenuate the neuronal activities within 4 mm diameter area. In addition, as shown in the Figure 4—figure supplement 1, the avian Wulst (A 8.00 to 14.50, 0 to L3.00, 0 to H3.00) in the pallium is a multilaminate structure which contains HA (hyperpallium apicale), HI (hyperpallium intercalatum), and HD (hyperpallium densocellulare). The projections of DLA mainly terminate in the HD and HI of the Wulst (Karten et al., 1973; Bagnoli and Burkhalter, 1983). The HA is the major area of Wulst that sends feedback projections to DLA, and forward projections to other visual and somatosensory/motor areas in the brain (Karten et al., 1973). Considering all these factors, we injected 1 µl muscimol (2%) at each of four different coordinates: (A10, L1.5, H1.5) and (A12, L1.5, H1.5) in the left hemisphere, (A10, L1.5, H1.5) and (A12, L1.5, H1.5) in the right hemisphere (Figure 4—figure supplement 1). The injection sites were also identified to locate within the HA.

As shown in Figure 4, we measured the oscillation frequency, duration of first replay cycle, and number of replay cycles of single replay cells before and after bilateral Wulst inactivation (n = 12 cells). We did not observe significant effects of Wulst inactivation on these parameters. In addition, without the feedback inputs from the pallial Wulst, the entrained oscillatory activities of replay cells sustained after LD stimulation and slowly declined over time. The oscillation frequencies of post-stimulus activities of replay cells linearly correlated with the temporal frequencies of LD stimuli before and after Wulst inactivation.

Experiment two: We do notice that the pharmacological inactivation of pallial Wulst in the experiment one had no significant effects on the post-stimulus oscillatory activities of DLA replay cells. To exclude the possible impact of incomplete Wulst inactivation, the electrolytic lesions were applied in the Wulst of both hemispheres. The lesion areas were confirmed from the brain slices (Figure 5—figure supplement 1).

As shown in Figure 5, DLA replay cells still sustained the entrained oscillatory activities after LD stimulation when the Wulst in both hemispheres was completely lesioned and the response characteristics of their post-stimulus activities were similar to those measured in the animals without Wulst lesion (Figure 2, Figure 3).

The results of additional experiments suggest that the feedback inputs from pallial Wulst do not contribute to the oscillatory activities of DLA cells under constant photic conditions after periodic stimulation. In addition to pallial inputs, the avian DLA also receives direct retinal inputs. However, the retinal cells responded to the omitted stimulus in repetitive stimuli with short intervals (20-100 ms) (Schwartz et al., 2007). When the stimuli in the seconds range were applied, the retinal cells did not show rhythmic activities after periodic visual stimulation (Sumbre et al., 2008). Taken these results together, the intrinsic circuits in the DLA rather than the retinal and pallial inputs play the main role in the entrained replay responses of DLA cells.

3) The model is too speculative. It either should be removed from the main body of the paper (and perhaps incorporated into the Discussion) or evidence should be provided to support its components.

Fully agree with the reviewers’ suggestions, we have moved the detailed description of computational model from the Results in the previous manuscript to the Materials and methods in the current version, and the computational model was only discussed in Discussion as Figure 6. The values of all parameters in the computational model were also listed in detail in the Materials and methods

As we stated in the Discussion of the manuscript, due to a variety of reasons, the present computational model was not fully conclusive. The lesion marks of recording sites only confirmed the recording area in the DLA but could not provide further information about the neuronal morphology and synaptic connections of recorded cells. The studies on the intrinsic electrical properties of DLA cells in the avian brain slices are also very few so far. Therefore, the computational model was only used to try to explain the potential neural mechanism underlying the experiment results. Further experiments, especially on the avian brain, are definitely required to fully understand the underlying neural mechanism.

Specifically: a) The oscillatory behavior is all captured in the "u" variable that has no clear connection to a biophysical property of calcium channels. Without additional measurements, this parameter ends up being simply a recapitulation of the phenomenon. Measurements of the particular conductances that cause the resonance (if that is even the source – it could be a network property) are beyond the scope of this study.

The I_Na+K+Ca_ model neuron (see Materials and methods) in the present experiment has been described and used in several previous studies (Morris and Lecar, 1981; Sherman and Rinzel, 1992; Izhikevich, 2007). By adjusting the parameters of model neuron, the neuron produces repetitive spiking or bursting. For example, the variable ‘S’ of model neuron in the Sherman and Rinzel’s work plays the same role as the variable ‘u’ in the present work. The authors suggested that the variable ‘S’ can be intracellular Ca^2+^ that slowly accumulates and activates a K^+^ current (Sherman et al., 1988).

The thalamocortical cells (TC cells) and interneurons in the mammalian LGN exhibit the voltage-dependent intrinsic oscillation (Zhu et al., 1999; Llinás and Steriade, 2006). In the brain slices, the properties of calcium channels and the contribution of I_Ca_ in the intrinsic oscillation of thalamic cells were carefully studied (Llinás and Jahnsen, 1982; McCormick and Pape, 1990; Soltesz et al., 1991). These works show that the interplay between the low-threshold Ca^2+^ (I_Ca_) and Na^+^-K^+^ current (I_Na+K_)/ non-selective cation current (I_CAN_) are crucial to the low-frequency oscillation (< 4 Hz) of TC cells or interneurons in the LGN.

Based on these studies, we proposed that the dynamic changes of I_Ca_ induced the oscillation responses of DLA cells to the periodic LD stimulus, which was reflected in the variable ‘u’ in the model neuron.

b) While inhibition from non-replay cells is one possible mechanism for the decay back to baseline, there is no experimental evidence for this, and (as the authors acknowledge in the Discussion), there are plenty of alternatives. Overall, the model seems like reasonable speculation for a Discussion section, but not a real result.

We designed this model primarily to answer two questions: why the DLA cells retained the entrained oscillatory activities after LD stimulation and why this post-stimulus oscillatory responses declined over time? The intrinsic voltage-dependent oscillation of thalamic cells might explain the first question. For the second question, the DLA in pigeons is abundant of GABA_A_, GABA_B_ and benzodiazepine binding sites (Veenman et al., 1994). Therefore, we introduced the inter-neuron inhibition into the model. In addition, we had ever tried different inter-neurons connections in the computational model, such as excitatory, inhibitory or combined excitatory and inhibitory. The simulation results with the inhibitory inter-neuron connection were consistent with the experimental results. As we expected, the oscillatory activities of model neuron after periodic stimulation attenuated under constant photic conditions over time. However, we cannot exclude other factors that might affect the persistent responses of thalamic cells in the absence of periodic inputs, such as the NMDA currents (Bottjer, 2005; Wang, 2001) and short-term synaptic plasticity in the neural network (Mongillo et al., 2008; Szatmáry and Izhikevich, 2010). In addition, it is known that interneurons can exert inhibitory influences on neurons projecting to the Wulst in the avian DLA (Miceli et al., 2008). It is plausible that the inhibitory inputs from the NR-neuron to the R-neuron in the model were mediated by inhibitory interneurons. In the present study, the computational model was simplified to only include two cells (R- neuron and NR-neuron). In the real brain, however, the post-stimulus oscillatory responses of replay cells are very likely to be generated by neural networks composed of a population of neurons in the DLA rather than a single cell (Gutnisky and Dragoi 2008; Wang et al., 2011; Benucci et al., 2013).

c) The asymmetric connection between R and NR neurons in the model is somewhat arbitrary. The authors should discuss anatomical justification of this connectivity based on known anatomical data.

By injecting tracer into the retina and Wulst in pigeons respectively, the fine structure of the DLA was analyzed using hodological techniques and GABA-immunocytochemistry (Miceli et al., 2008). As the mammalian LGN, the avian DLA receives direct retinal inputs and has reciprocal connections with the pallial Wulst. There are two types of GABA-immunonegative thalamopallial projection neurons and a single type of strongly GABA-immunoreactive interneurons in the DLA.

Two types of projection neurons are discriminated on the basis of cytological characters of neuronal somata. Type-1 projection neurons are ovoid in shape with major diameters of 12 μm and minor diameters of 7 μm. Type-2 projection neurons are round to fusiform with a major diameter of 14–16 μm and a minor diameter of 6–8 μm. Relative to type-1 projection neurons, type-2 projection neurons display well-developed Golgi cisterns and large Nissl bodies. In pigeons, 72% of the GABA-immunonegative retinal terminals contact projection neurons and 28% contact interneurons in the DLA. However, the intrinsic electrical properties of DLA projection neurons and their possible inter-neuron connections are unknown. Further experimental evidences are required to prove whether the projection neurons are the DLA replay/non-replay cells (R-/NR-neuron) and the possible relationship between them.

The retinothalamic and palliothalamic axon terminals are GABA-immunonegative in the DLA. In addition, the interneurons in the DLA exert inhibitory influences on neurons projecting to the Wulst. It is plausible that the inhibitory inputs from NR-neurons to R-neurons in the computational model were mediated by the inhibitory inter-neurons.

[Editors' note: further revisions were requested prior to acceptance, as described below.]

Reviewer #1:

In my view, the authors have done an excellent job addressing the reviews with additional experiments to test the robustness of the effect and whether it depends on feedback from Wulst. Moving the model to the Discussion helps to identify it as more a theory to inspire future experiments than an actual well-constrained result.

Thanks for the comments of the reviewer.

Reviewer #2:

Overall the authors addressed my concerns. However there are still a couple of more minor issues to consider:

Thanks for the comments of the reviewer.

The authors performed two series of experiments to investigate the possibility that the emergence of replay cells could be due to a feedback from the visual Wulst.· Pharmacological inactivation. This is thoroughly described in Materials and methods and in Figure 4—figure supplement 1. However they lack a positive control – an electrophysiological recording in HA showing that they can effectively shut down neuronal activity in this area. If this were available it would increase the value of these data.

A new experiment was conducted to confirm that the muscimol effectively inactivate the neuronal activities in Wulst. Both GABA_A_ and GABA_B_ binding sites are abundant in the avian Wulst (Veenman et al., 1994). To examine the spatial range and temporal course of muscimol inhibitory effect in Wulst cells, the neuronal activities of Wulst cells were determined by the mean firing rates of eight recording positions. Each recording site was 0.5 mm from the injection site. The electrode accessed each recording site one by one and continuously recorded 10 min of spontaneous activity at each site. It took about 1.5 hr to complete the recordings of all eight positions. We measured the activity changes of Wulst cells surrounding three different injection sites before and after injection. Our results are consistent with the previous works that the muscimol can effectively inhibit the neuronal activities in the brain for the long term (Partsalis et al., 1995; Arikan et al., 2002). The results were presented in Figure 4—figure supplement 3 and Results section.

· Surgical lesion. The procedure, animal recovery and lesioning parameters (e.g. current, duration) are not well described in the Materials and methods section, although Figure 5—figure supplement 1 clearly shows that their procedure is very effective. A bit more description in the Materials and methods would be helpful.

The method of pharmacological inactivation or electrolytic lesions of the Wulst in both hemispheres in details was described in the Materials and methods.

In comparison to other nuclei in the avian brain, the Wulst (A8.00 to 14.50, 0 to L3.00, 0 to H3.00) is relatively large area. To achieve the complete Wulst lesion in both hemispheres, we first tried to use the normal multi-site electrolytic lesions with the electrode by increasing the applied current and duration. However, from the brain slices of Wulst-lesioned animals, we found that this method was ineffective and could not guarantee the complete Wulst lesion. Then, we chose the hand-hold CH-HI Cautery (Advanced Meditech International, USA) to make the lesion. With CH-HI Cautery, the lesion surgery was done quickly and the bleeding in the surgery was effectively reduced. The brain slices of lesioned animals also confirmed the effectiveness of this method.

During the surgery, the physical conditions of anesthetized animals were strictly monitored, such as the animal’s breath, heart rate, body temperature. After the surgery, the lesion area was covered with the sterilized medical hemostatic sponge. The electrophysiological recordings started at one hour after the surgery.

The finding that both procedures do not have any effect on the ability of some DLA cells to replay periodic luminance stimuli is convincing. However, given the extent of these procedures, especially the electrolytic lesioning, I would expect to see some nonspecific effects such as a global reduction/increase in basal or evoked firing rates. As firing rates are normalized in order to compare peak responses (as in Figure 4, Figure 5) these nonspecific effects cannot be inferred from the data presented.

We analyzed the basal firing rates and photic responses of single DLA replay cell before and after muscimol injection in the Wulst of both hemispheres. The firing rates of replay cells under constant photic conditions before periodic stimulation were slightly reduced after pharmacological Wulst inactivation. The photic responses of DLA replay cells evoked by periodic LD stimuli were evaluated by a preference index of their photic responses (R) to light (L = 200 lux) and dark stimuli (D = 0 lux) (index = (R_L_ – R_D_) / (R_L_ + R_D_)). No significant change in response preference for photic stimuli was observed after Wulst inactivation.

The same method of analysis was also applied to compare the spontaneous activities and photic responses of the DLA replay cells recorded in normal and Wulst-lesioned animals. The Wulst lesion did not affect the spontaneous activities and photic responses of DLA replay cells. Under constant photic conditions before LD stimulation, no significant change in the spontaneous firing rate was observed between the replay cells recorded in the normal animals and those recorded in the Wulst-lesioned animals. Furthermore, the photic responses of DLA replay cells were also not affected by Wulst lesions, which was reflected in no significant change in the preference index for photic stimuli for both light-activated and light-suppressed

A panel with raw firing rates, like Figure 1 before and after pharmacological block and/or electrolytic lesions would be helpful in building the argument that periodic replay is selectively resistant to these modifications.

As suggested by reviewers, a typical replay cell with the raw firing traces before and after muscimol injection was presented in the new Figure 4—figure supplement 2.

In case no nonspecific effects can be observed the authors should provide some additional lines in Discussion explaining that outcome by providing plausible reasons as to why eliminating feedback from the visual Wulst had no impact on DLA activity in their recordings.

Based on the previous and present studies, we briefly discussed the possible reasons why the Wulst had no effect on the post-stimulus replay behavior of DLA cells in Discussion.

Given that neither pharmacological inactivation nor electrolytic lesions of the Wulst in both hemispheres affected the post-stimulus oscillatory activities of the DLA cells entrained by external, slow frequency LD periodic stimulation, the time-interval representation of avian thalamic cells in the order of minutes is unlikely to be modulated by the pallium Wulst. Like the mammalian LGN, the avian DLA is involved in far more than the simple transmission of visual information from the retina to the visual pallium. In addition to the retinal and pallial Wulst projections, the DLA also receives afferent supplies from the suprachiasmatic nucleus (SCN) and the optic tectum (Miceli et al., 2008; Cantwell and Cassone, 2006). However, we do not know which cognitive functions of visual thalamus are modulated by the SCN and tectum in the pigeon. Therefore, further neurophysiological evidence is required to reveal the possible modulating inputs to DLA cells in time-interval representation.